# History of incarceration and age-related neurodegeneration: Testing models of genetic and environmental risks in a longitudinal panel study of older adults

**Peter T. Tanksley** [1,2]*, **Matthew W. Logan** [3], **J. C. Barnes** [4]

1 Population Research Center, University of Texas at Austin, Austin, TX, United States of America,
2 Department of Psychology, University of Texas at Austin, Austin, TX, United States of America, 3 School of Criminal Justice and Criminology, Texas Status University, San Marcos, TX, United States of America,
4 School of Criminal Justice, University of Cincinnati, Cincinnati, OH, United States of America

* peter.tanksley@austin.utexas.edu

**Data Availability Statement:** Most of the data used in this study is publicly available from the HRS website: https://hrs.isr.umich.edu/. However, data on APOE genotypes are considered sensitive

## Abstract

History of incarceration is associated with an excess of morbidity and mortality. While the incarceration experience itself comes with substantive health risks (e.g., injury, psychological stress, exposure to infectious disease), most individuals eventually return from prison to the general population where they will be diagnosed with the same age-related conditions that drive mortality in the non-incarcerated population but at exaggerated rates. However, the interplay between history of incarceration as a risk factor and more traditional risk factors for age-related diseases (e.g., genetic risk factors) has not been studied. Here, we focus on cognitive impairment, a hallmark of neurodegenerative conditions like Alzheimer's disease, as an age-related state that may be uniquely impacted by the confluence of environmental stressors (e.g., incarceration) and genetic risk factors. Using data from the Health and Retirement Study, we found that incarceration and *APOE-ε4* genotype (i.e., the chief genetic risk factor for Alzheimer's disease) both constituted substantive risk factors for cognitive impairment in terms of overall risk and earlier onset. The observed effects were mutually independent, however, suggesting that the risk conveyed by incarceration and *APOE-ε4* genotype operate across different risk pathways. Our results have implications for the study of criminal-legal contact as a public health risk factor for age-related, neurodegenerative conditions.

## Introduction

The effects of incarceration are insidious, pervasive, and correspond with a litany of physical and mental health afflictions among the formerly incarcerated [1–7]. Although the causal mechanisms underlying these relationships are nuanced, there is considerable evidence to suggest that imprisonment exacerbates pre-existing medical conditions and other risk factors due to high rates of exposure to environmental stressors, such as prison violence and victimization,

and require the submission of a sensitive data access use agreement.

**Funding:** • The author(s) received no specific funding for this work.

**Competing interests:** The authors have declared that no competing interests exist.

and low rates of identification and treatment during the intake process. Research also indicates that the effects of incarceration are durable and extend into the communities to which individuals return, negatively impacting their quality of life and expectancy [8].

Of particular importance to practitioners and policymakers is the influence of the incarceration experience on acute and chronic causes of early mortality. In the time period around their release, people leaving prison are more likely to die from opioid use disorders (OUDs), overdose, suicide, and homicide [9–17]. One large-scale review on incarceration and health found that most post-release mortality studies focus on short-term effects and indicate that the period immediately following release represented a high-risk time for early mortality among formerly incarcerated persons due to acute causes [18]. The long-term impact of incarceration on mortality (i.e., via chronic health conditions) is less well understood. Most of the formerly incarcerated survive the high-mortality period immediately following release and eventually succumb to the same chronic conditions that drive mortality for the general population—albeit at exaggerated rates, relative to the general population [19], resulting in earlier mortality [20–22]. Examples of chronic conditions associated with past incarceration include cancer, cardiovascular disease, and neurodegenerative diseases such as Alzheimer's Disease and Related Dementias (ADRDs) [19, 23–26]. Our focus here is on the latter affliction; specifically, the potential long-term effects of incarceration on the onset and progression of neurodegeneration as indicated by cognitive impairment.

Relative to other chronic conditions, neurodegenerative diseases and their potential link with incarceration history have received less attention [27] (but see [28–30]). Still, existing research generally finds that age-related cognitive impairments (i.e., the chief indicator of neurodegeneration) are more prevalent among those who have been incarcerated at some point in their lives compared to the general population [23, 24, 26]. This is unsurprising, as formerly incarcerated persons are subject to a constellation of risk factors including acute stressors (e.g., traumatic brain injury, high infectious disease burden), chronic stressors (e.g., disenfranchisement, unemployment), and lack of social integration (e.g., difficulty maintaining familial connections, homelessness) [18, 31]. Together, these factors frame incarceration as a social determinant of health [32, 33], a non-medical factor impacting health such as health-related knowledge, attitudes, beliefs, and behaviors [34]. It has been noted that incarceration may, in fact, constitute both a cause and consequence of social determinants of health as many such factors that are exacerbated by the incarceration experience also raise the risk of initial incarceration (e.g., substance use) [35]. Here, we focus on the downstream consequences of incarceration as they relate to neurological health. And while it is observed that formerly incarcerated persons bear a heavier than expected burden of age-related neurodegeneration [24], it is not well-understood if/how the risks associated with the incarceration experience relates the other main source of risk for neurodegeneration: genetic risk factors.

Neurodegenerative diseases possess complex genetic etiologies. Alzheimer's disease (AD), for example, is oligogenic, meaning it is primarily influenced by a small number of genetic variants—including apolipoprotein E (*APOE*), a gene that codes for proteins that bind and transport low-density lipids and contribute to cholesterol clearance from the bloodstream [36]. Variation in this gene can impact cholesterol metabolism and may lead to increased risk for stroke, cardiovascular disease, and diagnosis of AD [37]. The *APOE* genotype is determined by two variants (rs7412 and rs429358), resulting in three main isoforms of protein *apoE*: E2, E3, and E4 encoded by the ε2, ε3, and ε4 alleles, respectively. The *APOE-ε4* allele has been linked with substantial increases in the risk of developing late-onset Alzheimer's disease. Possession of one copy of the *APOE-ε4* allele confers a 3-fold increase in risk while two copies confer a 15-fold increase, making *APOE-ε4* status the strongest genetic predictor of Alzheimer's disease risk [38]. Recent genome-wide association studies have begun to illuminate polygenic

variation that also contributes to AD [39], but highly impactful genes like *APOE* still loom large in the landscape of genetic risk.

Despite their strong genetic basis, neurodegenerative diseases are also affected by environmental risk factors—the likes of which correspond with the established literature on the social determinants of health within the domain of medical sociology. From this perspective, there exists a host of social, environmental, and economic risk factors/stressors that, if unaddressed, are detrimental to general medical and behavioral health and eventuate in the form of serious disorder among the general population. Indeed, the 2020 Lancet Commission report on dementia prevention, intervention, and care identified 12 modifiable risk factors that, together, accounted for 40% of dementias world-wide [40]. These included education, hearing loss, traumatic brain injury (TBI), hypertension, alcohol intake, obesity, smoking, depression, social isolation, physical inactivity, diabetes, and air pollution. Many of these risk factors are social determinants of health (e.g., alcohol intake, smoking, social isolation) and are typically the result of more "upstream" social determinants (e.g., socioeconomic position, education, criminal-legal involvement) [34].

With substantive environmental and genetic risk factors, neurodegeneration is highest among those for whom these risks co-occur, but it is unclear whether the risk posed by *APOE-ε4* positivity is additive or multiplicative. For instance, research on mouse and primate models suggest that the negative effects of the *APOE* gene network may become amplified in response to the kinds of nano-sized particulate matter in air pollution [41]. *APOE-ε4* positivity is also associated with poorer longer-term outcomes following TBI [42]. Thus, the risk of neurodegeneration posed by *APOE-ε4* positivity may be interactive with modifiable environmental risk factors raising the possibility of multiplicative effects. As incarceration is a locus of both acute (e.g., TBI) and chronic risk factors (e.g., social isolation), formerly incarcerated people with *APOE-ε4* positivity may have risk profiles that supersede that of either source of risk alone.

Here, we draw these two lines of research together to examine how incarceration history and *APOE-ε4* genotype combine to produce risk for neurodegeneration as indicated by cognitive impairment. We posit four possible scenarios (**Fig 1**): (1) genetic risks dominate prediction of cognitive impairment ("genetic risk model"); (2) environmental risks dominate prediction of cognitive impairment ("environmental risk model"), (3) genetic and environmental risks independently predict cognitive impairment ("G+E model"), and (4) genetic and environmental risks interact to produce multiplicative risk for cognitive impairment ("G×E model"). We test for evidence of these four scenarios in terms of both overall risk (i.e., ever developing cognitive impairment) and progression of cognitive impairment (i.e., age of onset).

## Methods

### Data

Data come from the Health and Retirement Study (HRS), a nationally representative cohort study of elderly Americans aged ≥50 [43]. The HRS collects information from participants on a biennial basis on social, demographic, economic, behavioral, and health conditions. In addition to the main survey, the HRS randomly selected half the sample to receive enhanced face-to-face interviews starting in 2006 and alternating to the other half of the sample with each wave of data collection. Enhanced interviews included physical performance measurements, blood and saliva samples, and a leave-behind psychosocial questionnaire. The current study relies on data on cognitive impairment from the main study, as well as *APOE-ε4* genotype and information on lifetime incarceration drawn from the enhanced interviews.

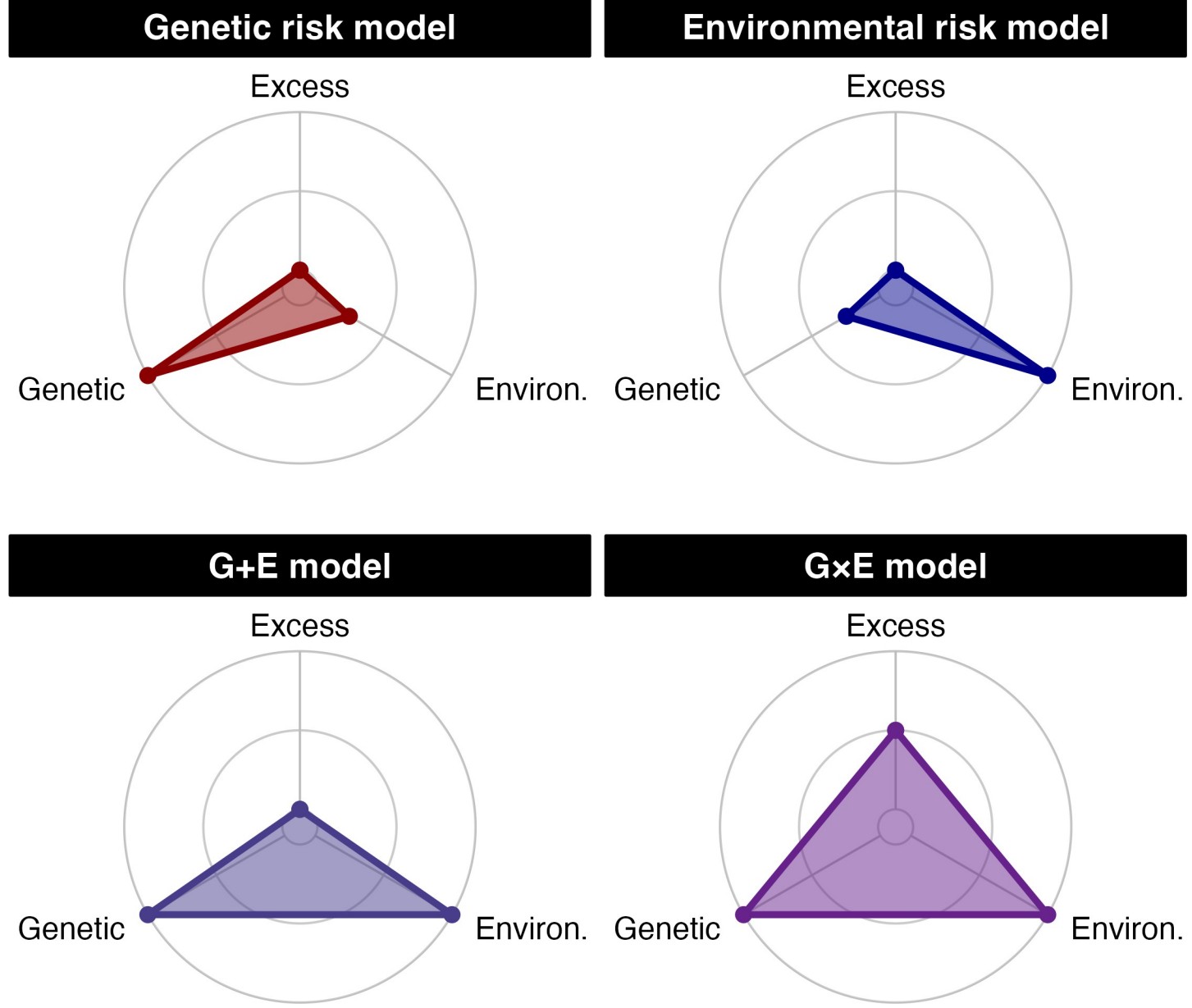

**Fig 1. Radar plots showing the theoretical models of prediction sources (genetic, environmental, excess) explored in the current study.** Distance from the origin of each plot represents the increasing magnitude of each risk factor's predictive power. Note: "Excess" refers to the additional prediction achieved by a model containing an interaction term.

## Measures

**Cognitive impairment.** We assessed cognitive status of HRS participants using the classification scheme developed for the HRS. All participants were administered the Telephone Interview for Cognitive Status (TICS), a telephone-based instrument that probed cognitive functioning along seven domains: memory, mental status, abstract reasoning, fluid reasoning, vocabulary, dementia, and numeracy. The TICS is a lightweight cognitive screening tool that is highly correlated with scores on alternative in-person assessments like the Mini-Mental State Examination ($r = 0.94$) and has high test-retest reliability (6-week $r_{test-retest} = 0.97$) [44]. In

2009, a methodology was developed for using TICS scores to classify respondents as "normal," "cognitively impaired non-dementia," and "demented." The classification system was validated by a team of dementia experts by comparing TICS scores to the Aging, Demographic, and Memory Study (ADAMS), a subsample of the HRS who received a more extensive psychological battery [45]. For the current analysis, we used repeated measures of the Langa-Weir Cognitive Status classifications for assessment years 2004–2020 from the HRS user contributed data set. To avoid small cell counts, we combined both non-normal cognitive states into one "impaired" category, resulting in a dichotomous *Cognitive impairment* variable.

**Lifetime incarceration.** HRS participants were asked about adverse experiences in adulthood including "Have you ever been an inmate in a jail, prison, juvenile detention center, or other correctional facility?" Respondents received a score of 1 if they answered "Yes" and a score of 0 if they answered "No." This question was asked in 2012/2014 and answers were coded so that if an individual answered "Yes" to either wave, they received a score of 1 on the *Lifetime incarceration* measure. Cases were only categorized as "missing" if they did not provide responses on both waves of data collection. The resulting variable indicated that approximately 8% of the analytic samples had some history of incarceration (**Table 1**). These observed rates of incarceration align with other national estimates [46]. We also created a variable that decomposed the *Lifetime incarceration* into periods of incarceration duration. For individuals who indicated that they had been incarcerated, they were asked a follow-up question regarding the lifetime duration of their time spent incarcerated with responses on a four-point Likert scale (less than one month, less than one year, between 1-5years, more than 5 years). We collapsed these categories into two durations (less than one more, a month or more) to avoid sparse response categories and assigned all participants who were never incarcerated a value of "none". The resulting three-category variable we termed *Lifetime incarceration duration*.

**APOE-ε4 genotype.** We assess monogenic risk for neurodegeneration by classifying HRS participants according to the number of *APOE-ε4* alleles they carry as the ε4 allele has been linked with large increases in risk for stroke and heart disease and is the leading genetic risk factor for Alzheimer's disease. *APOE* genotype was assessed in the HRS using two methods. The primary method relied on direct genotyping using TaqMan allelic discrimination SNP assays. Saliva samples for DNA extraction were collected from consenting HRS participants during years 2006, 2008, 2010, and 2012 [47]. Among individuals for whom direct genotyping was not conducted, *APOE* genotype was classified using imputed genotype array data. Following Ware and colleagues [48], we removed samples with imputed *APOE* genotypes with INFO Scores of <99%. We coded *APOE-ε4* genotype according to the number of ε4 alleles contained in each participants' genotype, ranging from 0–2, with more alleles representing greater risk.

**Covariates.** Every model included a "baseline" adjustment for age, self-reported sex (Male = 1, Female = 0), self-reported race/ethnicity, high school completion (Yes = 1, No = 0), and HRS birth cohort as covariates. We also estimated models with "full" adjustment using an extended panel of covariates that was designed to capture the leading modifiable environmental (non-genetic) risk factors for neurodegenerative diseases as outlined in the 2020 report from the Lancet's commission on dementia prevention, intervention, and care. This panel of environmental risk factors included: hypertension, hearing difficulty, smoking status, obesity, depression, physical (in)activity, diabetes, social isolation, alcohol consumption, and traumatic brain injury (note: air pollution exposure, one of the risk factors identified in the 2020 commission report, was not provided by HRS). To this panel, we also added childhood financial difficulty and household include to account for previous and contemporaneous differences in socioeconomic position, as well as stroke status. Full details regarding the measurement of covariates in the HRS and their missingness (**S1 Fig**) are provided in the **S1 Data**. The analytic sample with complete data included *N* = 11,268 cases with *N* = 73,511 observations.

**Table 1. Individual and observation-level descriptive statistics of the health and retirement study sample.**

| Individual-level | Overall (N = 11,268)[1] | Ever incarcerated? | | p-value[2] |
|---|---|---|---|---|
| | | No (N = 10,335)[1] | Yes (N = 933)[1] | |
| Lifetime incarceration duration | | | | – |
| None | 10,335 (92%) | 10,335 (100%) | – | |
| Less than one month | 610 (5.4%) | – | 610 (65%) | |
| One month or more | 323 (2.9%) | – | 323 (35%) | |
| *APOE-ε4* count | | | | 0.14 |
| Zero copies | 8,450 (75%) | 7,775 (75%) | 675 (72%) | |
| One copy | 2,595 (23%) | 2,356 (23%) | 239 (26%) | |
| Two copies | 223 (2.0%) | 204 (2.0%) | 19 (2.0%) | |
| Self-reported sex | | | | <**0.001** |
| Female | 6,747 (60%) | 6,509 (63%) | 238 (26%) | |
| Male | 4,521 (40%) | 3,826 (37%) | 695 (74%) | |
| Race/Ethnicity | | | | <**0.001** |
| White | 8,060 (72%) | 7,553 (73%) | 507 (54%) | |
| Black | 1,680 (15%) | 1,404 (14%) | 276 (30%) | |
| Hispanic | 1,217 (11%) | 1,092 (11%) | 125 (13%) | |
| Other | 311 (2.8%) | 286 (2.8%) | 25 (2.7%) | |
| HS completion | | | | <**0.001** |
| High school or more | 9,346 (83%) | 8,674 (84%) | 672 (72%) | |
| Less than high school | 1,922 (17%) | 1,661 (16%) | 261 (28%) | |
| Smoking status at baseline | | | | <**0.001** |
| Yes | 6,175 (55%) | 5,452 (53%) | 723 (77%) | |
| No | 5,093 (45%) | 4,883 (47%) | 210 (23%) | |
| Childhood financial difficulty | 0.90 (1.14) | 0.87 (1.12) | 1.23 (1.30) | <**0.001** |
| Childhood TBI | | | | <**0.001** |
| Yes | 1,089 (9.7%) | 946 (9.2%) | 143 (15%) | |
| No | 10,179 (90%) | 9,389 (91%) | 790 (85%) | |
| HRS cohort (birth year)[3] | | | | <**0.001** |
| AHEAD (<1924) | 498 (4.4%) | 479 (4.6%) | 19 (2.0%) | |
| CODA (1924–30) | 635 (5.6%) | 617 (6.0%) | 18 (1.9%) | |
| HRS (1931–41) | 4,483 (40%) | 4,226 (41%) | 257 (28%) | |
| WB (1942–47) | 1,294 (11%) | 1,207 (12%) | 87 (9.3%) | |
| EBB (1948–53) | 2,241 (20%) | 1,971 (19%) | 270 (29%) | |
| MBB (1954–59) | 2,117 (19%) | 1,835 (18%) | 282 (30%) | |
| **Observation-level** | **Overall** (N = 73,511)[1] | **Ever incarcerated?** | | **p-value**[2] |
| | | **No** (N = 68,220)[1] | **Yes** (N = 5,291)[1] | |
| Age | 68.62 (10.39) | 68.88 (10.40) | 65.28 (9.79) | <**0.001** |
| Cognitive function | | | | <**0.001** |
| Impaired | 11,348 (15%) | 10,172 (15%) | 1,176 (22%) | |
| Normal | 62,163 (85%) | 58,048 (85%) | 4,115 (78%) | |
| Household income (10K) | 7.11 (13.65) | 7.25 (13.97) | 5.32 (8.41) | <**0.001** |
| Alcohol (daily avg.) | 0.37 (0.86) | 0.34 (0.78) | 0.67 (1.55) | <**0.001** |
| Body-mass index | 29.03 (6.19) | 29.01 (6.19) | 29.32 (6.15) | 0.20 |
| Depressive symptoms | | | | <**0.001** |
| No symptoms | 36,586 (50%) | 34,637 (51%) | 1,949 (37%) | |
| 1–2 symptoms | 23,577 (32%) | 21,690 (32%) | 1,887 (36%) | |
| 3+ symptoms | 13,348 (18%) | 11,893 (17%) | 1,455 (27%) | |

(*Continued*)

**Table 1.** (Continued)

| Individual-level | Overall (N = 11,268)[1] | Ever incarcerated? | | p-value[2] |
|---|---|---|---|---|
| | | No (N = 10,335)[1] | Yes (N = 933)[1] | |
| Diabetes | | | | **0.008** |
| Never diagnosed | 57,002 (78%) | 53,181 (78%) | 3,821 (72%) | |
| Diagnosed | 16,509 (22%) | 15,039 (22%) | 1,470 (28%) | |
| Hearing difficulty | | | | **<0.001** |
| Normal | 59,119 (80%) | 55,268 (81%) | 3,851 (73%) | |
| Impaired | 14,392 (20%) | 12,952 (19%) | 1,440 (27%) | |
| Hypertension | | | | **<0.001** |
| Yes | 43,932 (60%) | 40,539 (59%) | 3,393 (64%) | |
| No | 29,579 (40%) | 27,681 (41%) | 1,898 (36%) | |
| Physical activity (light) | | | | **<0.001** |
| Every day | 7,448 (10%) | 7,022 (10%) | 426 (8.1%) | |
| >1 per week | 36,249 (49%) | 34,023 (50%) | 2,226 (42%) | |
| 1 per week | 18,370 (25%) | 17,050 (25%) | 1,320 (25%) | |
| 1–3 per month | 5,441 (7.4%) | 4,821 (7.1%) | 620 (12%) | |
| never | 6,003 (8.2%) | 5,304 (7.8%) | 699 (13%) | |
| Stroke status | | | | 0.228 |
| Yes | 5,186 (7.1%) | 4,701 (6.9%) | 485 (9.2%) | |
| No | 68,325 (93%) | 63,519 (93%) | 4,806 (91%) | |
| Social isolation | 2.37 (1.38) | 2.35 (1.38) | 2.67 (1.44) | **<0.001** |
| Study year | | | | **<0.001** |
| 2004 | 7,622 (10%) | 7,157 (10%) | 465 (8.8%) | |
| 2006 | 7,870 (11%) | 7,398 (11%) | 472 (8.9%) | |
| 2008 | 8,045 (11%) | 7,548 (11%) | 497 (9.4%) | |
| 2010 | 8,107 (11%) | 7,500 (11%) | 607 (11%) | |
| 2012 | 10,302 (14%) | 9,482 (14%) | 820 (15%) | |
| 2014 | 9,955 (14%) | 9,165 (13%) | 790 (15%) | |
| 2016 | 7,819 (11%) | 7,222 (11%) | 597 (11%) | |
| 2018 | 7,294 (9.9%) | 6,739 (9.9%) | 555 (10%) | |
| 2020 | 6,497 (8.8%) | 6,009 (8.8%) | 488 (9.2%) | |

[1] n (%); Mean (SD)

[2] P-values correspond to a Pearson Chi-squared test (APOE, sex, race/ethnicity, high school completion, smoking status, childhood TBI, HRS cohort) and a Wilcoxon rank sum test (childhood financial difficulty) comparing differences across incarceration groups.

[3] HRS cohorts: AHEAD = Asset and Health Dynamics among the Oldest Old; CODA = Children of the Depression; HRS = Health and Retirement Study (original); WB = War Babies; EBB = Early Baby Boomers; MBB = Mid Baby Boomers.

[1] Mean (SD); n (%)

[2] P-values correspond to a Pearson chi-square test (study year), t-test of person-level means (age, household hold income, alcohol intake, BMI, depressive symptoms, physical activity), and a repeated measures logistic regression (cognitive function, diabetes, hearing difficulty, hypertension, and stroke status) comparing differences across incarceration groups.

Note: it is possible that many of the risk factors included in our extended panel of covariates behave as mediators and/or confounders. For instance, problematic alcohol use could behave as both a confounder (incarceration ← alcohol → cognitive impairment), a mediator (incarceration → alcohol → cognitive impairment), or both (alcohol → incarceration → alcohol → cognitive impairment). As the measurement of lifetime incarceration in the HRS restricts our ability to establish temporal order among covariates and exposure to the incarceration

experience, our use of the extended panel of covariates is considered exploratory and we likewise restrict our primary focus to the models with the "baseline" adjustment where temporal order with incarceration experience is more certain and statistical overcontrol (i.e., controlling for mediators) is less likely [49].

## Analytic strategy

We estimated the overall risk for cognitive impairment in the HRS as a function of *Lifetime incarceration* and *APOE-ε4* genotype using a generalized mixed linear modeling framework. We estimated incidence rate ratios (IRRs) of cognitive impairment using Poisson regression models with a random intercept at the individual level to account for non-independence due to multiple observations. This approach removes interrelatedness among observations from the same individual, allowing for the interpretation of results as if they were drawn from a cross-sectional sample of unrelated individuals. Next, we estimated the hazard for onset of cognitive impairment using two time-to-event methods. First, using nonparametric Kaplan-Meier survival curves, we compared the unadjusted differences between strata of interest (e.g., incarcerated vs. never-incarcerated; *APOE-ε4* allele counts of zero vs. one. vs. two) using log-rank tests. Next, covariate adjustment was accomplished using Cox proportional hazard models. Time-varying covariates (e.g., stroke, physical activity) were accommodated by creating panel datasets with one row per individual per observation. Both survival methods used chronological age as the time-scale, an alternative to the time-on-study approach, because it is recommended for analyses in elderly cohorts with age-dependent outcomes [50, 51]. According to recommendations for age-scale survival models [52], we addressed possible calendar effects by stratifying Cox models by HRS birth cohort (i.e., allowing each birth cohort to have a different baseline risk). Additionally, as the age-based time-scale effectively accounts for age, we did not include age as a covariate in the Cox models. We also removed all participants from survival analyses that were showed signs of cognitive impairment at their baseline wave of data collection to prevent left-censoring ($N = 1,237$ participants were removed). The proportionality assumption was assessed for all Cox models by examining Schoenfeld residuals; we observed no evidence of violations.

Next, we performed two sets of subgroup analyses to better contextualize results. First, we decomposed our measure of *Lifetime incarceration* into two categories of total lifetime incarceration duration (i.e., less than one month; more than one month) and re-estimated our main models. This effectively tests for a possible dose-response relationship between lifetime incarceration and cognitive impairment. Second, we included interaction terms between our focal predictors and the variables of sex, race/ethnicity, and high school completion. This analysis allowed us to probe our results for heterogeneity across key demographic dimensions (**Table 3**). Note: as both subgroup analyses involved potential interaction terms between categorical variables, we ensured adequate statistical power and reduced the risk of violating the anonymity of participants by establishing a minimum case count of twenty unique cases for all subgroupings of crossed variables. Based on this criterion, we did not fit any three-way interactions (e.g., *APOE-ε4×Lifetime incarceration×Race/ethnicity*) or any interactions involving *Lifetime incarceration duration*. We also observed that some categories of race/ethnicity did not meet this criterion in our sample. Rather than foregoing this analysis, we removed individuals categorized as either Hispanic or "other" and estimated a separate model comparing Black/White participants across our focal predictors (e.g., *Race×APOE-ε4*, *Race×Lifetime incarceration*).

## Results

### Do genetic risk factors predict cognitive impairment in older adults?

We found that *APOE-ε4 genotype* was associated with an increased risk of impaired cognitive function during the study period (**Table 2**). Consistent with prior research, we observed that, relative to those with no *APOE-ε4* alleles, risk for cognitive impairment increased as *APOE-ε4* allele count increased from one allele (IRR = 1.24, 95% CI [1.15–1.34], P<0.001) to two (IRR = 1.57, 95% CI [1.26–1.96], P<0.001) (**Model 2.1**). When estimated in the same model as *Lifetime incarceration* (**Model 2.3**) the results were essentially unchanged. Full adjustment for the extended panel of environmental risk factors had similarly negligible impact (**Model 2.4**) suggesting that the monogenic risk for cognitive impairment conferred by *APOE-ε4 genotype* is independent of *Lifetime incarceration* and key environmental risks.

### Is lifetime incarceration a risk factor for cognitive impairment in older adults?

We found that *Lifetime incarceration* conferred a substantive increase in the risk of cognitive impairment (**Table 2**). HRS participants with a history of incarceration were 1.5 times as likely to be cognitively impaired than their non-incarcerated peers (IRR = 1.45, 95% CI = [1.29, 1.62], P<0.001) (**Model 2.2**). Substantively, this effect size falls between the risk of having one or two *APOE-ε4* alleles (e.g., IRRs = 1.24–1.57). As before, we found that association between *Lifetime Incarceration* and *Cognitive Impairment* were virtually unchanged when *APOE-ε4 genotype* was included in the model (**Model 2.3**). Although we did observe a reduction in effect size in the fully adjusted model (IRR = 1.26 95% CI = [1.13, 1.40], P<0.001; **Model 2.4**). Despite the reduction in effect size (∼20%), *Lifetime incarceration* still represented a source of substantive risk for cognitive impairment in the sample that is independent of most leading risk factors (environmental and genetic) for neurodegenerative disease.

These findings support the interpretation that *APOE-ε4 genotype* and *Lifetime incarceration* operate as independent risk factors for cognitive impairment in later adulthood. The possibility remains, however, that these genetic and environmental sources of risk, when brought

**Table 2. Mixed effect Poisson regression of cognitive status on *APOE-ε4* genotype and lifetime incarceration in the HRS ($N_{Observation}$ = 73,511; $N_{Cases}$ = 11,268).**

| Variable[1] | Model 2.1 (baseline adjustment) | | Model 2.2 (baseline adjustment) | | Model 2.3 (baseline adjustment) | | Model 2.4 (full adjustment) | | Model 2.5 (baseline adjustment) | | Model 2.6 (full adjustment) | |
|---|---|---|---|---|---|---|---|---|---|---|---|---|
| | IRR[2,3] | 95% CI[3] | IRR[2,3] | 95% CI[3] | IRR[2,3] | 95% CI[3] | IRR[2,3] | 95% CI[3] | IRR[2,3] | 95% CI[3] | IRR[2,3] | 95% CI[3] |
| *APOE-ε4* allele count | | | | | | | | | | | | |
| one copy | 1.24*** | [1.15, 1.34] | | | 1.23*** | [1.15, 1.33] | 1.23*** | [1.14, 1.32] | 1.25*** | [1.15, 1.35] | 1.24*** | [1.15, 1.33] |
| two copies | 1.57*** | [1.26, 1.96] | | | 1.58*** | [1.27, 1.97] | 1.54*** | [1.25, 1.89] | 1.67*** | [1.33, 2.10] | 1.62*** | [1.31, 2.00] |
| Lifetime incarceration | | | 1.45*** | [1.29, 1.62] | 1.44*** | [1.29, 1.62] | 1.26*** | [1.13, 1.40] | 1.50*** | [1.31, 1.71] | 1.30*** | [1.15, 1.48] |
| Lifetime Incarceration | | | | | | | | | | | | |
| × One copy | | | | | | | | | 0.91 | [0.71, 1.17] | 0.91 | [0.73, 1.14] |
| × Two copies | | | | | | | | | 0.51 | [0.23, 1.16] | 0.55 | [0.25, 1.17] |
| (Intercept) | 0.04*** | [0.03, 0.04] | 0.04*** | [0.03, 0.04] | 0.04*** | [0.03, 0.04] | 0.03*** | [0.03, 0.04] | 0.04*** | [0.03, 0.04] | 0.03*** | [0.03, 0.04] |

[1] The "baseline" adjustment for all models included age, sex, race/ethnicity, high school completion, and HRS cohort. Models with "full" adjustment (models 2.4, 2.6) also adjusted for stroke status, alcohol intake, BMI, depression symptoms, diabetes status, hearing difficulty, hypertension, household income, (light) physical activity level, smoking history, social isolation, childhood financial hardship, and childhood traumatic brain injury.

[2] *p<0.05; **p<0.01; ***p<0.001

[3] IRR = Incidence Rate Ratio, CI = Confidence Interval

**Table 3. Subgroup analysis of the of mixed effects Poisson models of cognitive status on *APOE-ε4* genotype and lifetime incarceration across dimensions of race, sex, and education.**

| Variable[1] | Model 3.1[4] (baseline adjustment) | | Model 3.2[4] (baseline adjustment) | | Model 3.3 (baseline adjustment) | | Model 3.4 (baseline adjustment) | | Model 3.5 (baseline adjustment) | |
|---|---|---|---|---|---|---|---|---|---|---|
| | IRR[2,3] | 95% CI[3] | IRR[2,3] | 95% CI[3] | IRR[2,3] | 95% CI[3] | IRR[2,3] | 95% CI[3] | IRR[2,3] | 95% CI[3] |
| *APOE-ε4* allele count | | | | | | | | | | |
| One copy | 1.25*** | [1.16, 1.36] | 1.30*** | [1.18, 1.43] | 1.23*** | [1.15, 1.33] | 1.33*** | [1.20, 1.46] | 1.27*** | [1.17, 1.39] |
| Two copies | 1.53*** | [1.20, 1.94] | 1.71*** | [1.28, 2.29] | 1.58*** | [1.27, 1.97] | 1.83*** | [1.37, 2.43] | 1.58*** | [1.23, 2.03] |
| Lifetime incarceration | 1.58*** | [1.39, 1.79] | 1.72*** | [1.46, 2.02] | 1.44*** | [1.29, 1.62] | 1.77*** | [1.43, 2.18] | 1.52*** | [1.32, 1.75] |
| Race/ethnicity (Black) | 3.06*** | [2.79, 3.35] | 3.31*** | [2.95, 3.71] | | | | | | |
| Sex (male) | | | | | 1.11** | [1.04, 1.19] | 1.19*** | [1.10, 1.29] | | |
| High school completion (no) | | | | | 3.06*** | [2.83, 3.31] | | | 3.20*** | [2.92, 3.51] |
| Race/ethnicity (Black) | | | | | | | | | | |
| × Incarcerated | | | 0.79 | [0.62, 1.00] | | | | | | |
| × One copy | | | 0.87 | [0.73, 1.05] | | | | | | |
| × Two copies | | | 0.71 | [0.44, 1.15] | | | | | | |
| Sex (male) | | | | | | | | | | |
| × Incarcerated | | | | | | | 0.76* | [0.59, 0.97] | | |
| × One copy | | | | | | | 0.84* | [0.72, 0.98] | | |
| × Two copies | | | | | | | 0.71 | [0.46, 1.11] | | |
| High school completion (no) | | | | | | | | | | |
| × Incarcerated | | | | | | | | | 0.87 | [0.69, 1.09] |
| × One copy | | | | | | | | | 0.88 | [0.74, 1.05] |
| × Two copies | | | | | | | | | 1.00 | [0.60, 1.68] |
| (Intercept) | 0.03*** | [0.03, 0.04] | 0.03*** | [0.03, 0.04] | 0.04*** | [0.03, 0.04] | 0.03*** | [0.03, 0.04] | 0.04*** | [0.03, 0.04] |
| $N_{Observations}$ | 64,955 | | 64,955 | | 73,511 | | 73,511 | | 73,511 | |
| $N_{Cases}$ | 9,740 | | 9,740 | | 11,268 | | 11,268 | | 11,268 | |

[1] All models also adjusted for age, sex, race/ethnicity, high school completion, and HRS cohort.

[2] *p<0.05; **p<0.01; ***p<0.001

[3] IRR = Incidence Rate Ratio, CI = Confidence Interval

[4] Due to a limited number of unique cases (<20) within categories of race/ethnicity across *Lifetime incarceration* and/or *APOE-ε4 genotype*, participants who self-identified as Hispanic or "other" ($N_{Cases}$ = 1,528) were removed. Multiplicative interaction terms test for differences between participants who identified as Black or White.

together, will go beyond their additive effects to inflict a multiplicative increase in risk for impairment.

### Does the incarceration experience amplify genetic risk factors for cognitive impairment in older adults?

We found no evidence of an interaction between *Lifetime incarceration* and *APOE-ε4 genotype* on the risk for cognitive impairment. Prior to formally testing for statistical interaction, we observed that *APOE-ε4 genotype* did not differ across participants with/without prior incarceration experience, suggesting that *APOE-ε4 genotype* did not contribute to selection into the carceral system which is a crucial assumption of G×E models (i.e., no gene-environment correlation) [53, 54]. We entered multiplicative interaction terms for *Lifetime incarceration* and *APOE-ε4 genotype* into our analysis (**Model 2.5**) and found that neither term (i.e., for one copy or two copies) predicted excess risk of cognitive impairment that was distinguishable from the additive effects. **Model 2.6** presents similar results with full covariate adjustment. Re-

estimating **Models 2.5** & **2.6** as a linear model under the linear probability framework produced the same null results (results not presented), suggesting the lack of interaction was not due to the scale of the interaction (i.e., multiplicative vs. additive) [55]. This finding supports the interpretation that *Lifetime incarceration* and *APOE-ε4 genotype* convey their risk for later life cognitive impairment in a fashion that is independent and additive in nature.

### Do genetic risk factors for AD predict earlier cognitive impairment?

In addition to conferring more risk for cognitive impairment at any point (reported above), we found that *APOE-ε4 genotype* was associated with risk for earlier onset of cognitive impairment as well. We estimated univariate Kaplan-Meier survival curves in our analytic sample ($N_{Person-years}$ = 117,142) using age at first indication of cognitive impairment and found statistically significant differences across *APOE-ε4 genotype* according to a log-rank test ($\chi^2(2)$ = 99.25, P<0.001) (**Fig 2, Panel A**). The ages of median survival probability across *APOE-ε4* genotypes were 84 (zero copies of *APOE-ε4*), 81 (one copy of *APOE-ε4*), and 77 (two copies of *APOE-ε4*). Baseline covariate adjusted was achieved with a Cox proportional hazard model (**S1 Table**). We observed that possession of one (HR = 1.25, 95% CI [1.16–1.35], P<0.001) or two copies (HR = 1.68, 95% CI [1.36–2.09], P<0.001) of *APOE-ε4* resulted in significantly greater hazard for developing cognitive impairment in later life (**Model S1.1**). As before, inclusion of *Lifetime incarceration* and additional covariates did not substantively change the effect estimates (**Models S1.3–4**).

### Does lifetime incarceration predict earlier cognitive impairment?

We found evidence that HRS participants with a history of incarceration were at risk for earlier cognitive impairment compared to those without a history of incarceration. Comparing Kaplan-Meier survival curves, we found that individuals with a history of incarceration tended to develop cognitive impairment earlier than their never-incarcerated peers (log-rank $\chi^2(1)$ = 255.74, P<0.001) (**Fig 2, Panel B**). There was an eight-year difference between the groups in terms of median survival probability (76 vs. 84). That is, the age at which half of HRS participants were estimated to have experienced cognitive impairment came almost a decade earlier for the group with a history of incarceration compared to those without such history.

A Cox proportional hazard model confirmed that *Lifetime incarceration* was associated with earlier cognitive impairment independent of baseline covariates (HR = 1.40, 95% CI [1.25–1.58], P<0.001) (**S1 Table**, **Model S1.2**). While the inclusion of *APOE-ε4* genotype in the model did not affect the estimated hazard ratio for lifetime incarceration (**Model S1.3**), the fully adjusted model did demonstrate a reduction in effect size (HR = 1.27, 95% CI = [1.13, 1.43], P<0.001; **Model S1.4**). Similarly, the addition of multiplicative terms did not indicate the presence of unexplained excess risk for participants who experienced incarceration and had one or more *APOE-ε4* alleles (**Model S1.5–6**).

### Subgroup analysis

We investigated possible heterogeneity within our results by decomposing *Lifetime incarceration* into a three-category measure of *Lifetime incarceration duration* (i.e., none, less than a month, a month or more). By separating transient exposures from more serious ones, we were able to probe for a possible dose-response relationship between incarceration and cognitive impairment. Re-estimating our main models, we found evidence for a dose-response relationship with risk of cognitive impairment increasing between participants with less than a month spent incarcerated (IRR = 1.36, 95% CI = [1.18, 1.56], P<0.001) and those who spent a month or more (IRR = 1.61, 95% CI = [1.35, 1.93], P<0.001) (**S2 Table**, **Model S2.1**). Full covariate

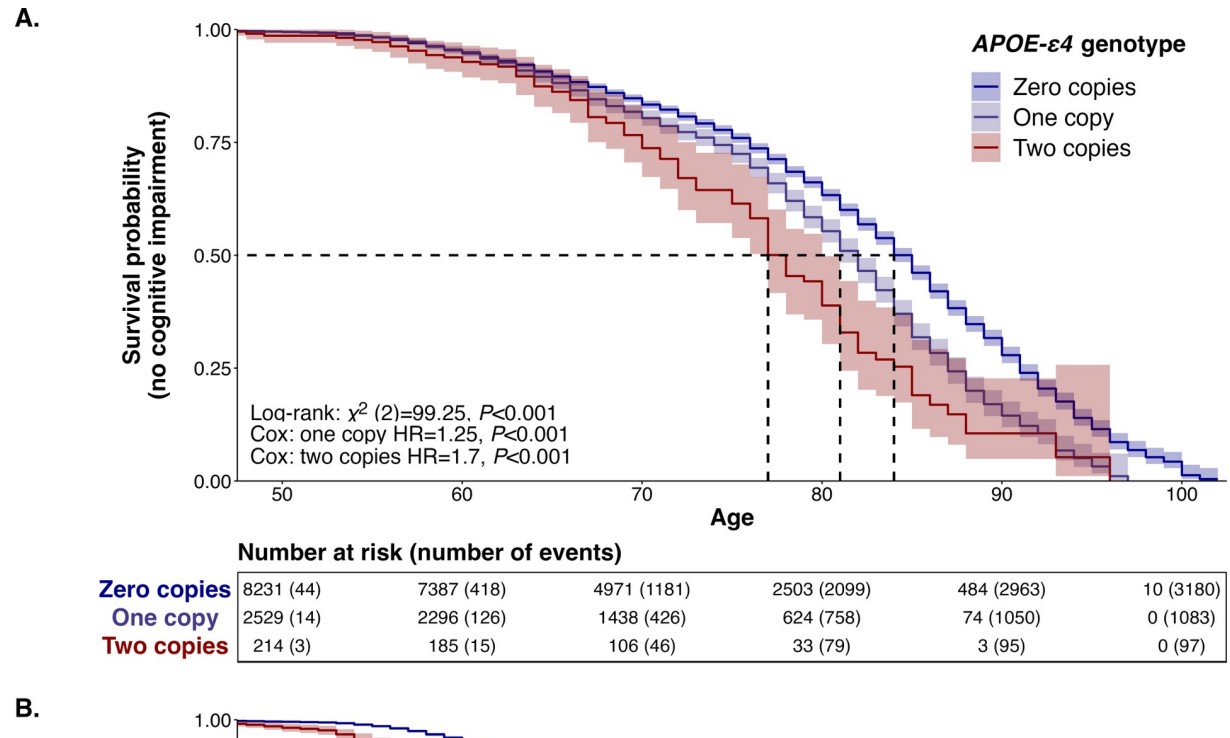

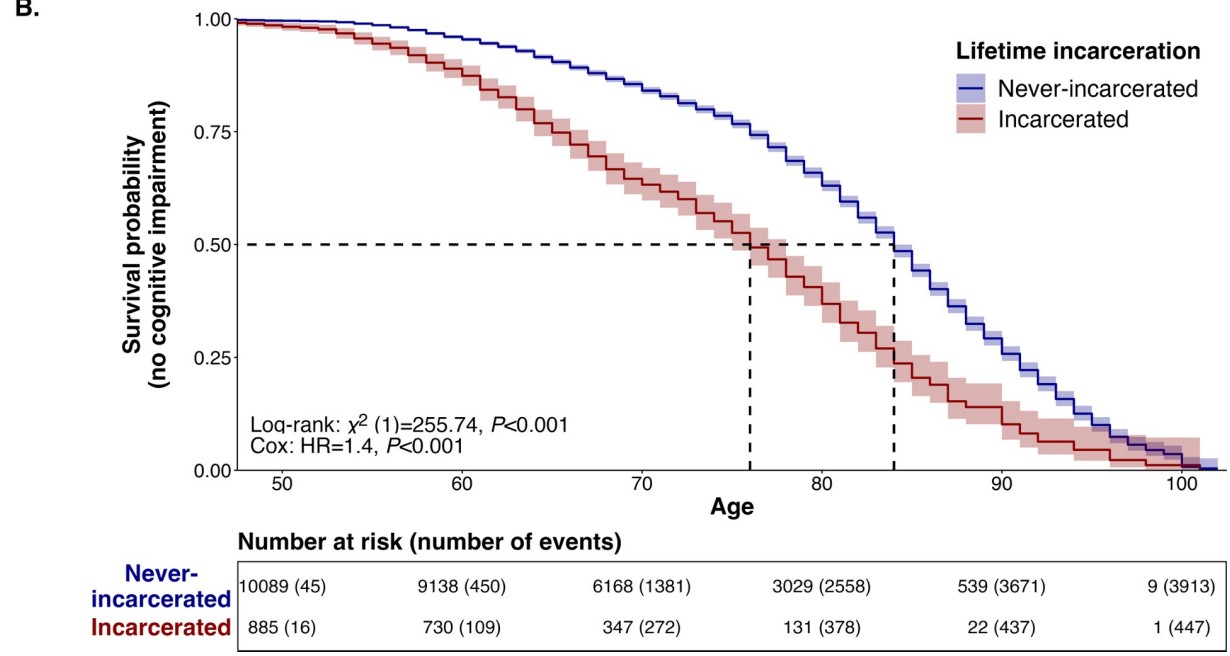

**Fig 2. Time-to-event models of *APOE-ε4* genotype and lifetime incarceration predicting the hazard of cognitive impairment among HRS participants.** Non-parametric Kaplan-Meier survival curves and risk tables for onset of cognitive impairment, stratified by *APOE-ε4* genotype (**Panel A**) and lifetime incarceration (**Panel B**), respectively. Time-scale corresponds to chronological age. Dashed lines indicate age at median survival probability per stratum. Number of events (i.e., first designation of cognitive impairment) are cumulative. Note: individuals were considered censored for any non-event status (e.g., death, non-participation); x-axis used a floor of age 50 for clarity.

adjustment reduced effect sizes by a marginal amount (IRRs ranged from 1.22–1.32), but it did not reduce statistical significance or change the pattern of increasing risk with incarceration duration (**Model S2.2**). These patterns of dose-response associations were again observed

when re-estimating the Cox models for the hazard of cognitive impairment as a function of *Lifetime incarceration duration* (**S3 Table**, **Models S3.1–2**).

Next, we tested for heterogeneity across the key demographic dimensions of race/ethnicity, sex, and education (i.e., high school completion). When comparing Black and White HRS participants, we found no evidence of group differences in terms of overall associations between *APOE-ε4 genotype* and *Lifetime incarceration* with cognitive impairment (**Table 3**, **Models 3.1–2**). We did find evidence for sex differences across associations with both *Lifetime incarceration* (IRR = 0.76, 95% CI = [0.59, 0.97], P = 0.025) and possessing a single *APOE-ε4* allele (IRR = 0.84, 95% CI = [0.72, 0.98], P = 0.024), suggesting that males who were previously incarcerated or had a single *APOE-ε4* allele experienced a lower risk level for cognitive impairment relative to females with the same status/background (**Models 3.3–4**). These findings accord with prior findings of a bias favoring males in terms of age-related neurodegeneration [56], the genetic risk conferred by *APOE-ε4* genotype [57], as well as the experience of collateral consequences of past incarceration [58]. Finally, we did not find evidence of group differences across groups based on early educational attainment (i.e., high school completion) (**Models 3.3** & **3.5**). Results of the subgroup analysis for the hazard for developing cognitive impairment mirrored those reported above, including the observed sex differences (**S4 Table**).

## Discussion

Cognitive decline is a hallmark of aging and a prerequisite for neurodegenerative diseases like Alzheimer's disease. The progression toward cognitive impairment and eventually dementia can be exacerbated by factors both genetic and environmental. Here, we examined two such risk factors–one well-known (*APOE-ε4* genotype) and one less well-known (incarceration history)–and sought to determine whether they function as independent or contingent risks. We observed that *APOE-ε4* genotype and lifetime incarceration were independently related with an increased risk for cognitive impairment among HRS participants (i.e., a "G+E model" of risk). Surprisingly, when estimating overall risk for cognitive impairment, lifetime incarceration was associated with a risk level that fell between the risk associated with carrying one and two copies of the *APOE-ε4* allele—the strongest genetic risk factor for Alzheimer's disease. We found also that people with a history of incarceration developed cognitive impairment almost a full decade earlier than people who did not have a history of incarceration.

These findings position lifetime incarceration as an environmental risk factor for cognitive impairment that is comparable to and independent of *APOE-ε4* genotype. However, we did observe some heterogeneity in our results as well. As might be expected, we found that people with more transient incarceration experiences (i.e., < one month lifetime duration) had a higher risk for cognitive impairment than those with none, but those with longer exposures (i.e., a month or greater) had the highest risk for cognitive impairment. These findings underline the heterogenous nature of the incarceration experience and the need for improved precision of its measurement in population health studies like the HRS. Future scholarship could substantively contribute to public health efforts by investigating the mechanisms that underly this dose-response pattern, be they direct (i.e., increased exposures during incarceration) or indirect ones (e.g., more social isolation arising from increased stigma of longer jail/prison time).

While we did not observe group differences in the risks conveyed by *APOE-ε4* genotype or past incarceration in terms of race (i.e., Black vs. White) or educational attainment (high school completion), we observed excess risks experienced by female participants compared to males. Women who had one copy of the *APOE-ε4* allele or who were previously incarcerated had greater risk for cognitive impairment than men with the same background/status. These findings align with prior findings identifying a male-favoring bias in terms of age-related

neurodegeneration [56], the impact of *APOE-ε4* positivity [57], and health-related collateral consequences of incarceration [58]. The presence of sex differences in the risk associated with both *APOE-ε4* genotype and incarceration suggests that our primary question related to the multiplicative effects of these factors on cognitive impairment might be restricted to women only—a possibility we could not test in the present population-based sample.

The above comparison of risks/hazards between *APOE-ε4* genotype and lifetime incarceration should be interpreted with caution as *APOE-ε4* genotype is a single, precisely measured risk factor and past incarceration is likely a proxy (i.e., not causal) for the many exposures that go along with a criminal lifestyle—although our analysis did adjust for lifestyle factors associated with both past incarceration and neurodegenerative diseases (e.g., alcohol consumption, social isolation). Nonetheless, we believe that criminal-legal contact, particularly at the level of incarceration, is an important indicator to consider in population health research for three reasons. First, past incarceration identifies a segment of the population that is disproportionately poor, low in academic achievement, and hailing from communities of color [59–63] all of which are risk factors for both neurodegenerative diseases and future contact with the carceral system. Even if treated merely as an indicator of the concentration of risk, past incarceration would be valuable information for research aiming to characterize the health burden in a population.

Second, consistent with observations made within the field of medical sociology [64] and the corresponding literature on the social determinants of health, there is evidence that past incarceration is not merely an indicator of the underlying risk of the (highly selected) carceral population, but also causally contributes to the health burden of the formerly incarcerated via direct (i.e., exposures during their sentence) or indirect means (i.e., incarceration-related exposures and collateral consequences experienced after release) [65–67]. Although this line of research is still nascent, it is instructive to consider given its implications for the reentry process, especially among historically vulnerable demographic groups [20, 31, 65].

Third, incarceration may exacerbate underlying risks (e.g., genetic risk factors) in a multiplicative fashion, going beyond the mere additive effects. The current study did not find evidence for the latter possibility (i.e., no statistical interaction between past incarceration and *APOE-ε4* genotype was observed) but we believe that such a scenario is worth examining in the case of other outcomes, especially age-related conditions with different genetic profiles (e.g., polygenic instead of oligogenic).

The current study relied on participants in a population health survey. This approach assumes that the population of interest (i.e., those affected by the incarceration experience) are likely to participate in population health surveys after their return to the general population. To understand the impact of incarceration on cognitive outcomes more fully (and the progression of age-related conditions more generally), we suggest that researchers move their attention to the incarceration experience itself. Our results comport with a litany of studies underscoring the necessity of improving risk assessment and classification for incarcerated persons, especially during the intake process (i.e., before entry into prison). Traditional assessments, such as the Level of Service Inventory-Revised (LSI-R), typically focus on indicators of criminogenic risk including antisocial cognitions and personality patterns and have been empirically validated over time and space for both men and women [68–70]. More recently, scholars have emphasized the importance of embedding indicators of psychiatric and medical afflictions into these tools as a means of improving service provision [71, 72]. There have also been calls to provide in-depth assessments and evaluations of the aging process within prisons [73]. It is hoped that the proactive application of such assessments will lead to more targeted provision of healthcare services within prisons/jails, as well as help elucidate the causes

processes underlying the incarceration-neurodegeneration nexus (i.e., how variation in the prison experience translates to within-person changes in cognitive impairment).

Early identification of risk for rapid physical and cognitive aging is particularly important for prisons due to the heightened age-related health burden already apparent in the carceral population. According to the United States Census Bureau, the threshold for identifying older persons among community samples is approximately 65 years old; by contrast, the National Institute of Corrections generally classifies anyone over the age of 50 as "older" [74, 75]. The most recent estimates and projections, though dated, indicate that incarcerated persons over the age of 50 maintain health profiles equivalent to persons outside of prison who are 65 and older [76]. Given the results of the current study, administrators might consider specialized or alternative placements for individuals with higher levels of genetic risk. Intake assessments could include *APOE-ε4* genotype as an additional tool for anticipating some of the financial and logistical encumbrances (e.g., healthcare spending and facility layout/design) that accompany the management of older incarcerated persons [77]. Such information could also be used to inform future decisions such as compassionate release. Of course, such considerations must be balanced against other indicators of risk, including the history of and propensity for violence [78], as well as the ability to maintain and provide safety and security to others. Incorporating knowledge of genetic risk for age-related morbidity into prison administration is particularly important because manifestations of genetic risk do not remain static over time; rather, they can be curbed or exacerbated, depending on the prison experience.

Disentangling the incarceration-neurodegeneration nexus will become of critical importance in the coming decades, as both the number of individuals living with ADRDs in the community and the "graying" of the prison population are projected to increase substantially. For example, recent estimates indicate that approximately 6 million individuals in the U.S. are currently living with ADRDS; by 2060, this figure will have more than doubled and directly impact nearly 14 million people [27]. By the same token, as of 2020, approximately 20% of the U.S. prison population met the age threshold to be considered an older person (≥50 years old); by 2030, this figure will rise to approximately 33% [25, 60]. To help address these grim projections, we believe future efforts should consider genetic risk factors and within-person change in cognitive function inside the prison context itself.

The results of the current study and aforementioned projections comport with suggestions made recently by Testa *et al.* [27] who emphasized the necessity of studying the nexus between incarceration and ADRDs throughout the life course because of its potential impact on (1) the reentry and reintegration of affected individuals, noting that it may increase social and economic disparities which independently correlate with general wellbeing; (2) the efficacy of treatment from caregivers and community service providers after an individual's release; and (3) our understanding of the causal ordering associated with incarceration, ADRDs, and life expectancy (i.e., causation vs. selection and reverse causality) [79, 80]. The status of incarceration as cause or consequence (or both) of social determinants of health is still under study. In the current study, we heeded Testa and colleagues' (2023) call for research by examining the enduring effects of imprisonment on cognitive impairment, a key characteristic of ADRD diagnosis.

The results and limitations of the current study offer several avenues for future research. First, the HRS did not collect information regarding the timing of incarceration, which prevented us from establishing temporal order. The lack of temporal order precludes our ability to examine many relevant aspects of the association between incarceration and cognitive decline. For instance, it is impossible to establish whether a third variable, like drug use, acts as a mediator or a confounder. To this end, the data do not permit use to test for reverse causality, and specifically the extent to which cognitive reserves and impairment affect the probability of

incarceration. As Testa *et al.* note, metrics of impulsivity, intelligence, and brain injury independently correlate with incarceration as well as cognitive decline. Future research should therefore consider these and other relevant factors when studying the incarceration-cognitive impairment nexus to parse out causal pathways. Second, the measurement of lifetime incarceration in the HRS combines many different forms of incarceration into a single indicator, likely concealing a large degree of heterogeneity in the carceral experience. The incarceration experience (i.e., the "dose") [27] varies across many dimensions including but not limited to the duration (which we examined), time period, age of onset, chronicity, security level, and subjective experience. It should therefore be a public health priority to understand which dimensions impart the greatest risk for cognitive decline, thus allowing more targeted policy conversations. Finally, participants of population health studies such as the HRS are not always representative specific sub-populations of interest to researchers (e.g., the formerly incarcerated population). It is possible that the current results were affected by this form of ascertainment bias, especially if those with incarceration experience and *APOE-ε4* risk alleles were disproportionately less likely to participate in the study. We do not believe this specific limitation to have greatly impacted our results, however, given that no difference in rates of *APOE-ε4* genotype between incarcerated/non-incarcerated groups were present in our analytic sample (**Table 1**).

## Conclusion

Incarceration is a heterogenous exposure with many downstream consequences for later life, some of which put formerly incarcerated persons at increased risk for cognitive decline. We found that incarceration imparted risk for cognitive impairment in later life that was independent of and comparable to the leading genetic risk factor (*APOE-ε4*). As a risk factor, past incarceration is potent but also opaque. More exploration of the mechanistic processes underlying the incarceration-cognitive impairment nexus may contribute to the amelioration of population-wide disparities in age-related conditions.

## Supporting information

**S1 Table. Cox proportional hazard model of lifetime incarceration and *APOE-ε4* genotype on first cognitive impairment in the HRS ($N_{Person-years}$ = 117,142; $N_{Cases}$ = 10,031).** Lifetime incarceration predicts hazard of cognitive impairment independent of *APOE-ε4* genotype and additional covariates.
(DOCX)

**S2 Table. Mixed effect Poisson regression of cognitive status on *APOE-ε4* genotype and lifetime incarceration duration in the HRS ($N_{Observation}$ = 73,511; $N_{Cases}$ = 11,268).** Lifetime incarceration predicts risk of cognitive impairment in a dose-response pattern.
(DOCX)

**S3 Table. Cox proportional hazard model of first cognitive impairment on *APOE-ε4* genotype and lifetime incarceration duration in the HRS ($N_{Person-years}$ = 117,142; $N_{Cases}$ = 10,031).** Lifetime incarceration predicts hazard of cognitive impairment in a dose-response pattern.
(DOCX)

**S4 Table. Cox proportional hazard model of first cognitive impairment on *APOE-ε4* genotype and lifetime incarceration across dimensions of race, sex, and education.** *APOE-ε4* genotype and Lifetime incarceration predicts risk of cognitive impairment differentially across sex, but no differences were detected across race (Black vs. White) or educational attainment

(high school completion).
(DOCX)

**S1 Fig. Missingness (%) across study variables in the full HRS study sample, stratified by missing status of the main lifetime incarceration question. Panel A** presents time-invariant variables (those items with fewer than $N = 80$ missing cases were not displayed). Childhood financial difficulty did not have any missingness among individual with valid lifetime incarceration data. Contrariwise, lifetime incarceration duration, being a follow-up question for the main question about lifetime incarceration duration, had only a single case with missing data on the main incarceration question. **Panel B** presents the time-varying variables by year of data collection. The social isolation index, as noted in the **S1 Data**, had particularly high missingness. This was due, in part, to the items contributing to index that were taken from the enhanced face-to-face interview that occurred every four years for participants.
(DOCX)

**S1 File. Sensitive data access use agreement.**
(PDF)

**S1 Data.**
(DOCX)

## Acknowledgments

We are grateful to the participants and team members of the Health and Retirement Study. The Health and Retirement Study is sponsored by the National Institute on Aging (grant number NIA U01AG009740) and is conducted by the University of Michigan.

## Author Contributions

**Conceptualization:** Peter T. Tanksley, Matthew W. Logan, J. C. Barnes.

**Data curation:** Peter T. Tanksley.

**Formal analysis:** Peter T. Tanksley.

**Visualization:** Peter T. Tanksley.

**Writing – original draft:** Peter T. Tanksley, Matthew W. Logan, J. C. Barnes.

**Writing – review & editing:** Peter T. Tanksley, Matthew W. Logan, J. C. Barnes.

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
