## [Decision Letter · Decision Letter 0]

22 Aug 2023

PONE-D-23-19488History of incarceration and age-related cognitive impairment: Testing models of genetic and environmental risk in longitudinal panel study of older adults.PLOS ONE

Dear Dr. Tanksley,

Thank you for submitting your manuscript to PLOS ONE. After careful consideration, we feel that it has merit but does not fully meet PLOS ONE’s publication criteria as it currently stands. Therefore, we invite you to submit a revised version of the manuscript that addresses the points raised during the review process.

We look forward to receiving your revised manuscript.

Kind regards,

Andrea Knittel

Academic Editor

PLOS ONE

Journal Requirements:

5. We notice that your supplementary figure and table are included in the manuscript file. Please remove them and upload them with the file type 'Supporting Information'. Please ensure that each Supporting Information file has a legend listed in the manuscript after the references list.

Additional Editor Comments:

Please note, in particular, the reviewer's suggestion to use person-first, non-stigmatizing language throughout the manuscript. Many authors have moved away from the terms “correctional” and “criminal justice” or “justice-involved” given the limited amount of correcting and justice that is present in the current system. Please consider using “carceral” when discussing facilities and systems that incarcerate people and “criminal legal” or “criminal legal involvement” when referring to the larger system of policies, policing, courts, etc. Person-first language is also an important step toward addressing the stigma of incarceration. Person-first language requires that the words “person” or “people” come first in the sentence. Person-first alternatives to stigmatizing labels such as "inmate" or "offender" would include “people with incarceration experience” or “people with criminal-legal involvement.”

Reviewers' comments:

Reviewer's Responses to Questions

**Comments to the Author**

1. Is the manuscript technically sound, and do the data support the conclusions?

Reviewer #1: Yes

Reviewer #2: Yes

Reviewer #3: Yes

2. Has the statistical analysis been performed appropriately and rigorously? 

Reviewer #1: Yes

Reviewer #2: Yes

Reviewer #3: I Don't Know

3. Have the authors made all data underlying the findings in their manuscript fully available?

Reviewer #1: Yes

Reviewer #2: Yes

Reviewer #3: Yes

4. Is the manuscript presented in an intelligible fashion and written in standard English?

Reviewer #1: Yes

Reviewer #2: Yes

Reviewer #3: Yes

5. Review Comments to the Author

Reviewer #1: Here is the full review:

My review was very favorable and recommend acceptance with minor edits.

This was an interesting and important study, that utilized data from the Health and Retirement Study, to examine the separate and combined effects of history of incarceration and APOE-ε4 genotype on cognition. The study was well -designed and the manuscript was clearly written, however some considerations for the authors are listed below.

1. Provide reference for TICS (Line 120-125)

2. What is the reliability for self-report of stroke? Provide references

3. In table 1, provide definitions for HRS cohort in footnotes

4. Were sex and race stratified analyses conducted? If not, would explore differences by sex and race since incarceration history differed significantly by sex and race, and the relationships between cognition and incarceration may also differ according to these factors

5. Are any data available on substance use and substance use history? Persons with substance use may be more likely to be incarcerated and also develop cognitive impairment and this should be accounted for.

6. As noted, the absence of data on time spend incarcerated, number of incarceration events, and most recent incarceration is a significant limitation

Reviewer #2: Authors address an important public health issue that is only recently gaining more attention in the literature – the potential association between incarceration and cognitive impairment. The authors sought to determine if incarceration and APOE-E4 genotype have independent or multiplicative associations with cognitive impairment as identified via the TICS. They used Health and Retirement Study data as HRS is one of few longitudinal datasets of older individuals that includes information on incarceration history.

There are some concerns that should be addressed:

1. Throughout the manuscript, the authors use terms such as inmates and offenders. There is a push to get away from this type of derogatory language and use more person-centered/humaninzing language such as “persons who have been incarcerated” or “persons convicted of a crime.” Relatedly, the authors also use the term elderly in the manuscript. This term should be replaced with language such as “older persons” or “older people.” Moreover, because the study includes those age 50 and older, the authors may consider referring to participants as those in mid to late life.

2. It is unclear from the supplementary figure how many individuals were missing information on incarceration? It would be helpful to know if those who are missing data on incarceration are also more likely to be missing data on cognition and genetic factors.

3. On page 5 (lines 88-90), the authors mention 12 modifiable factors that have been linked to dementia. Many of these variables are included in the HRS data (e.g., depression, alcohol, physical inactivity, diabetes, hypertension). These variables have also been found to be associated with incarceration. It is unclear why the authors mention this important potential confounders yet they do not control for these in the adjusted models, particularly when they control for variables such as stroke and social origins index? The authors should provide some information regarding why certain variables were or were not selected to be included in the adjusted models. It is also highly recommended that the authors repeat the analyses controlling for these additional variables as time-varying covariates.

4. Related to Comment #3, on page 16 (lines 253-255) the authors indicate that incarceration confers the same risk for cognitive impairment as APOE-E4 count. Yet, without controlling for factors such as depression and alcohol, this finding may be overstated.

5. Per an article cited by the authors (Garcia-Grossman), the HRS contains information on time spent incarcerated. So, it is unclear why the authors did not control for this variable as they mention on page 20 that duration of incarceration may impact outcomes.

6. Based on the findings in Table 1, there was nearly the same distribution of APOE-E4 count in those with and without a history of incarceration. Based on the similar distribution, it does not seem surprising that an interaction term between incarceration and APOE-E4 would be non-significant. Could the authors comment further in the Discussion on the added value of evaluating this interaction?

7. On page 17 (lines 281-284) and page 18 (lines 302-304) the authors discuss risk classification for cognitive status at the time of intake. However, for older persons who spent a considerable amount of time incarcerated, classification at the time of intake may be very different than their risk classification during the extent of their incarceration. The points made on these 2 pages should likely be discussed together. The authors should clarify how risk classification at the time of intake may or may not impact future cognitive status/services related to cognition.

8. Citation #62 is more than 20 years old. Do the authors have a more recent citation that provides evidence that incarcerated persons age 50 are similar to non-incarcerated persons age 65?

9. It is well known that incarceration is more prevalent among minority groups in the U.S. At some point, the authors should acknowledge the vast differences in the proportion of incarceration within the different race groups in the HRS sample. Although the majority of those incarcerated were white, if you look within race categories there is a very different picture: 7.6% whites incarcerated; 18.9% blacks incarcerated; and 11.5% Hispanics incarcerated. Did the authors evaluate an interaction between race and incarceration or race APOE-E4 count?

Reviewer #3: This is an interesting article that contributes to the field's understanding of the relationship of incarceration on dementia and cognitive decline. My comments mostly involve wording, language and phrasing to help better communicate the ideas and reduce stigma. INTRO - Overall, this section needs to be strengthened to better document that people who are incarcerated generally have worse health than the GP (see the Bureau of Justice Statistics health briefs for data) because they generally experience more hardships in the community (housing insecurity, exposure to violence, poverty, etc) that put them at risk for incarceration AND poor health. These same factors then overlap with risks for dementia and incarceration exacerbates risks for dementia. I suggest incorporating the social determinants of health model (SDOH) to tie things together. Please avoid "offenders" and "inmates" where possible. Although these are still commonly used in the criminal justice system, the academic field and health field have moved away from using these terms. "Most former inmates survive the high-mortality period immediately following release and eventually succumb to the same chronic conditions that drive mortality for the general population such as cancer, cardiovascular disease, and neurodegenerative diseases." This sentence needs to be rethought, while not factually incorrect, it obscures the point that people with CJ generally have worse health than GP, and buries that people with a history of incarceration are likely going to die from those conditions earlier than the GP. “Minoritized groups” is odd phrasing. And it's not the race that is the risk factor, it's the racism they experience, so please be careful with how that is worded.

The discussion does a better job of tying ideas together, but it also could use some strengthening to better connect the ideas. Again, I think the SDOH model will be helpful framing.

6. PLOS authors have the option to publish the peer review history of their article (what does this mean?). If published, this will include your full peer review and any attached files.

Reviewer #1: No

Reviewer #2: No

Reviewer #3: No

---

## [Author Response · Author response to Decision Letter 0]

15 Oct 2023

We thank the Editor and the Reviewers for their time and consideration. We look forward to your assessment of the improvements we have made on our article in response to your valuable feedback.

---

## [Decision Letter · Decision Letter 1]

20 Nov 2023

History of incarceration and age-related neurodegeneration: Testing models of genetic and environmental risks in a longitudinal panel study of older adults

PONE-D-23-19488R1

Dear Dr. Tanksley,

We’re pleased to inform you that your manuscript has been judged scientifically suitable for publication and will be formally accepted for publication once it meets all outstanding technical requirements.

Kind regards,

Andrea Knittel

Academic Editor

PLOS ONE

Additional Editor Comments (optional):

Reviewers' comments:

Reviewer's Responses to Questions

**Comments to the Author**

1. If the authors have adequately addressed your comments raised in a previous round of review and you feel that this manuscript is now acceptable for publication, you may indicate that here to bypass the “Comments to the Author” section, enter your conflict of interest statement in the “Confidential to Editor” section, and submit your "Accept" recommendation.

Reviewer #1: All comments have been addressed

2. Is the manuscript technically sound, and do the data support the conclusions?

Reviewer #1: Yes

3. Has the statistical analysis been performed appropriately and rigorously? 

Reviewer #1: Yes

4. Have the authors made all data underlying the findings in their manuscript fully available?

Reviewer #1: Yes

5. Is the manuscript presented in an intelligible fashion and written in standard English?

Reviewer #1: Yes

6. Review Comments to the Author

Reviewer #1: (No Response)

7. PLOS authors have the option to publish the peer review history of their article (what does this mean?). If published, this will include your full peer review and any attached files.

Reviewer #1: No

---

## [Editor Report · Acceptance letter]

24 Nov 2023

PONE-D-23-19488R1 

History of incarceration and age-related neurodegeneration: Testing models of genetic and environmental risks in a longitudinal panel study of older adults. 

Dear Dr. Tanksley:

I'm pleased to inform you that your manuscript has been deemed suitable for publication in PLOS ONE. Congratulations! Your manuscript is now with our production department. 

Kind regards, 

on behalf of

Dr. Andrea Knittel 

Academic Editor

PLOS ONE